# Soil Fungal Community Differences in Manual Plantation Larch Forest and Natural Larch Forest in Northeast China

**DOI:** 10.3390/microorganisms12071322

**Published:** 2024-06-28

**Authors:** Mingyu Wang, Xin Sui, Xin Wang, Xianbang Zhang, Xiannan Zeng

**Affiliations:** 1Engineering Research Center of Agricultural Microbiology Technology, Ministry of Education & Heilongjiang Provincial Key Laboratory of Ecological Restoration and Resource Utilization for Cold Region & Key Laboratory of Microbiology, College of Heilongjiang Province & School of Life Sciences, Heilongjiang University, Harbin 150080, China; wmy022234@163.com; 2Heilongjiang Zhongyangzhan Black—Billed Capercaillie National Nature Reserve Service Center, Nenjiang 161400, China; zh_xi_ba@163.com; 3Institute of Crop Cultivation and Tillage, Heilongjiang Academy of Agricultural Sciences, Harbin 150088, China; zengxiannanzxn@163.com

**Keywords:** manual plantation larch forest, natural larch forest, soil fungal communities, co-occurrence network, community assembly

## Abstract

Soil fungal communities are pivotal components in ecosystems and play an essential role in global biogeochemical cycles. In this study, we determined the fungal communities of a natural larch forest and a manual plantation larch forest in Heilongjiang Zhongyangzhan Black-billed Capercaillie Nature Reserve and Gala Mountain Forest using high-throughput sequencing. The interactions between soil fungal communities were analysed utilising a co-occurrence network. The relationship between soil nutrients and soil fungal communities was determined with the help of Mantel analysis and a correlation heatmap. The Kruskal–Wallis test indicated that different genera of fungi differed in the two forest types. The results show that there was a significant change in the alpha diversity of soil fungal communities in both forests. In contrast, nonmetric multidimensional scaling (NMDS) analysis showed significant differences in the soil fungal community structures between the manual plantation larch forest and the natural larch forest. The soil fungal co-occurrence network showed that the complexity of the soil fungal communities in the manual plantation larch forest decreased significantly compared to those in the natural larch forest. A Mantel analysis revealed a correlation between the soil fungal co-occurrence network, the composition of soil fungi, and soil nutrients. The RDA analysis also showed that AN, TK, and pH mainly influenced the soil fungal community. The null model test results showed the importance of stochastic processes in soil fungal community assembly in manual plantation larch forests. Overall, this study enhances our understanding of the differences in soil fungal communities in manual plantation larch forests and natural larch forests, providing insights into their sustainable management. It also serves as a reminder that the ecological balance of natural ecosystems is difficult to restore through human intervention, so we need to protect natural ecosystems.

## 1. Introduction

Forest ecosystems play a crucial role in the biosphere, not only absorbing large amounts of carbon dioxide, but also playing a very important ecological role in soil and water conservation [1,2,3,4]. In addition, larch forests play a pivotal role in maintaining ecological balance and promoting biodiversity [5,6,7]. However, in recent years, environmental pollution caused by human activities, such as deforestation, large-scale grazing, and carbon dioxide emissions, has led to the destruction of the natural larch forest ecosystem [8,9,10]. Intensifying the study of artificial larch forests holds substantial importance in preserving biodiversity, fostering ecological rehabilitation, and advancing sustainable progress [11,12,13]. Especially in Northeast China, larch forests are not only an important part of the forest ecosystem, but also play an indispensable role in soil protection, water conservation and climate regulation in the region. Further study of larch forests, especially planted larch forests, is essential for maintaining and improving the quality of the ecological environment in Northeast China.

Soil fungi are an integral part of forest ecosystems, and they play a key role in organic matter decomposition, nutrient cycling, soil structure formation and symbiosis with plants [14]. Changes in soil fungal communities directly affect the health and stability of forest ecosystems [15]. These tiny organisms construct a complex subsurface ecological network through various interactions such as competition, reciprocity and predation, which have a profound impact on forest carbon cycling, nutrient dynamics and even the functioning of the whole ecosystem [16,17]. In addition, soil fungi in natural and planted forests show significant differences in nutrient cycling and plant symbiosis [18]. Fungi in natural forests are rich in species and form reciprocal relationships with a variety of organisms through a complex below-ground ecological network, effectively decompose organic matter and promote nutrient release [19], are strongly associated with soil organic matter and specific enzymes, and thus efficiently participate in nutrient cycling and establish a long-term and stable symbiotic relationship with plants to enhance their environmental adaptability and resistance [20,21]. In contrast, fungal communities in planted forests are more closely related to specific tree species and may be more efficient in cycling certain nutrients under artificial management, but the symbiotic relationship with plants is more dependent on human management and may be less diverse and stable than in natural forests due to the homogeneity of plant species [22,23]. These differences mainly stem from the differences in ecological environment, plant diversity and anthropogenic management between the two forest stands. In the context of biodiversity conservation and ecological restoration, the study of soil fungal communities is of great significance. By exploring soil fungi in depth, we can improve our knowledge of soil biodiversity and identify and conserve those fungal species that are critical to the ecosystem. At the same time, this research can contribute to the overall conservation of biodiversity.

An in-depth understanding of the complex interactions among soil fungi, such as competition, reciprocity and predation, is particularly important to study community assembly processes [24,25]. As for soil fungal community assembly, it is not only related to the construction of soil fungal communities, but also closely related to many ecosystem functions [26]. Meanwhile, soil fungal co-occurrence network analysis, as an effective research method, can help us to explore these differences in depth and reveal the interrelationships among different fungal species and their ways of coexisting and interacting in the ecosystem [27,28]. Given this background, investigating the community assembly process of soil fungi and its co-occurrence network in manual plantation larch forest and natural larch forest is crucial for understanding the construction mechanism and ecological adaptations of fungal communities, which not only helps to reveal the functions and interactions of soil fungi in ecosystems, but also provides a powerful guide for biodiversity conservation and ecological restoration [29].

This study was conducted in the representative ecosystem of Zhongyangzhan Black-billed Capercaillie Nature Reserve and Gala Mountain Forest, located in the northeast of Heilongjiang Province, China. Due to the ecological significance of the region, this study involved the analysis of the differences in the soil fungal communities in a manual plantation larch forest as well as a natural larch forest. We used high-throughput sequencing technology to investigate the soil fungal community composition, structure, symbiotic network, and community assembly process in the two forest types. We hypothesise the following: (1) There may be some differences in the soil nutrient content in different forest types (natural larch forest and manual plantation larch forest). (2) Soil fungal communities’ co-occurrence network patterns have different complexities in different forest types (natural larch forest and manual plantation larch forest). (3) The balance between deterministic and stochastic processes in the soil fungal community assembly may differ in different forest types (natural larch forest and manual plantation larch forest).

## 2. Materials and Methods

### 2.1. Study Area

The two study areas were the natural larch forest located at the Zhongyangzhan Black-billed Capercaillie Nature Reserve (126°00′–126°45′ N, 48°30′–48°50′ E) and the manual plantation larch forest located in Gala Mountain Forest (50°47′25″–50°59′30″ N, 125°58′50″–126°20′12″ E) (Figure 1). We chose territories with superior natural conditions and ecological environments as natural forest sampling sites. In planted forests, we used natural succession and ecological restoration to minimise human intervention and increase ecosystem stability and diversity. The natural larch forest is located in the transition zone between the southwestern foothills of the Xiaoxinganling Mountains and the Songnen Plain, with a total area of 7274 ha. The region belongs to the temperate continental monsoon climate zone, with long, cold winters and short, cool summers, with a frost-free period of 121 d and an average annual rainfall of 450–550 mm. The mean temperature is −0.4 °C, with a maximum temperature of 37 °C and a minimum temperature of −48 °C. The natural larch forest contains 50% *Larix gmelinii*, 30% *Quercus mongolica*, 10% *Betula dahurica*, and 10% *Betula platyphylla*. The vegetation density is 42 plants/1000 m^2^, the closure degree is 0.6, the average diameter at breast height (DBH) is 28 cm, and the average tree height is 21 m. The manual plantation larch forest has a cold temperate continental monsoon climate with cold, dry winters and warm, rainy summers. The average annual temperature in the manual plantation larch forest is about −3.5 °C, the frost-free period is approximately 80 days, the average annual rainfall is 490 mm, and the effective cumulative temperature is about 1900 °C. The manual plantation larch forest contains 100% *Larix gmelinii*. The plantation forest consists of 100% larch, with a density of 85 plants/1000 m^2^, a closure of 0.8, an average diameter at breast height (DBH) of 18 cm, and an average tree height of 24 m.

### 2.2. Soil Sampling

Soil samples were acquired in July 2019 from the selected plots in the natural larch and manual plantation larch forests. Within each plot, after removing leaves and dry vegetation, soil samples (0–20 cm depth) were collected along an S-shaped path using a soil auger with a diameter of 8 cm. Then, the collected soil samples were mixed. A total of 12 soil samples were collected. Soil samples were taken from each of the two forests, and then placed in a refrigerator. We set up 6 replicates during soil collection (6 replicate samples in each of the planted and natural forests). Next, soil samples were transported to the laboratory and divided into two portions; one portion was air-dried and thoroughly ground for soil chemical analyses, and the other portion was stored at −80 °C for soil fungal DNA extraction.

### 2.3. Characterisation of Soil Chemical Parameters

The soil chemical properties were characterised as described in our previous study [30]. Briefly, the soil–water (deionised water) (1:2.5 *w*/*v*) suspension was shaken for 30 min, and then the pH was measured using a pH meter (Thermo Scientific Orion 3-Star Benchtop, Cambridge, United Kingdom) [31]. The total N (TN) and total K (TK) were quantified using an elemental analyser (Elementar, Langenselbold, Germany) [32]. The measurement of available nitrogen (AN) was more complex and was sequentially processed in H_2_SO_4_-HClO_4_, 0.5 M NaHCO_3_, and 2.0 M KCl, followed by assaying using a continuous flow analyser system (SKALAR SAN^++^, Breda, The Netherlands) [30]. The total phosphorus (TP) was measured through wet digestion with HClO_4_-H_2_SO_4_ followed by measurement using a spectrophotometer [30]. We quantified the available phosphorous (AP) using 0.5 M NaHCO_3_ extractive colourimetry [30]. The available potassium (AK) was measured through NH_4_OAc extraction [33].

### 2.4. DNA Extraction, PCR Amplification, and MiSeq Sequencing

We used the Fast DNA SPIN Extraction Kit (MP Biomedicals; Santa Ana, CA, USA) to extract genomic DNA from 0.5 g of soil according to the instructions. The quantity of DNA was detected using agarose gel electrophoresis (1%) and assayed using a NanoDrop ND-1000 spectrophotometer (Thermo Fisher Scientific; Waltham, MA, USA). The fungal ITS rRNA region was amplified using primers ITS1 (5′-CTTGGTCATTTAGAGGAAGTAA-3′) and ITS2 (5′-GCTGCGTTCATCGATGC-3′). A 7 bp sample-specific DNA barcode was added to the primers of each unique sample for multiplex sequencing. PCR amplification was then performed, and the PCR amplification conditions were as follows: The PCR reaction mixture volume was 25 μL and consisted of 12.5 μL PCR Master Mix (TAKARA, Shiga, Japan), 2.0 μL forward and reverse primers (5 μmol L^−1^), 2.0 μL DNA, and 6.5 μL H_2_O. The PCR amplification procedures included an initial denaturation step at 95 °C for 45 s, followed by 30 cycles at 95 °C for 45 s, 55 °C for 50 s, and 72 °C for 60 s, with a final extension step at 72 °C for 10 min. Each PCR was performed in triplicate. PCR products were purified using the TAKARA DNA Gel Extraction Kit (TAKARA Biosciences) [30]. The purified PCR products were pooled in equal proportions and were quantified using Qubit 3.0 (Life Invitrogen, Waltham, MA, USA). During DNA extraction, we used detergents and salts for efficient cell lysis and DNA release, followed by purification via phenol–chloroform, high salt precipitation, or column methods to achieve high purity. For sequencing, we conducted rigorous quality checks on DNA samples, employing high-throughput technologies like Illumina and PacBio for precise sequencing. We also repeated sequencing and validated results to promptly identify and rectify any errors, thus ensuring data accuracy.

### 2.5. Analysis of Sequencing Data

We used the QIIME1 [34] to analyse the raw readings. UPARSE was used for constructing OTUs de novo from next-generation reads [35]. Taxonomic classification was carried out using the Unite 8.0 Fungal Database. The BLAST algorithm was then utilised via QIIME on sequences from the UNITE 8.0 database [35]. The abundance of each operational taxonomic unit (OTU) in each sample and the taxonomy of these OTUs were then tabulated. The alpha diversity indices (e.g., Observed_species index, Chao1 index, ACE index, Shannon index, and Simpson index) were calculated in QIIME1 using the previously mentioned OTU table.

### 2.6. Statistical Analyses

One-way analysis of variance (ANOVA) was conducted to identify variations in the soil chemical properties and fungal alpha diversity, after which least significant difference tests were conducted. We performed nonmetric multidimensional scaling (NMDS) based on Bray–Curtis matrices using the “vegan” package in the R software to analyse the soil fungal communities’ β-diversity. Soil fungal community indicator species taxa were identified using linear discriminant analysis (LDA) effect size (Lefse) analysis. We used “microeco” and “ggplot2” packages in the R software to build a Lefse graph. Random forest analyses were carried out using the “random Forest”, “psych”, “reshape2” and “ggplot2” packages in R software [36] to identify the key soil fungal taxa that play an important role in plantation forest conversion. The random forest model was constructed based on the decision tree classification algorithm. Evolutionary trees (heat trees) for species classification were plotted using the “metacoder” package in R [37]. We used Gephi (http://gephi.github.io/, accessed on 1 June 2024) to draw co-occurrence network visualisations. In addition, we used the subgraph function in the “igraph” package in the R software to select sub-networks of co-occurrence networks from the global network and divide them according to the given nodes [38,39,40]. The Mantel test was conducted using the “vegan” package in R [41]. We analysed the relationship between soil fungal co-occurrence networks, the composition of soil fungi, and soil chemical properties. The null model calculates the Beta Nearest Taxon Index (βNTI) using the “Picante” package in R to elucidate whether deterministic or stochastic processes dominate the community assembly process [42]. The Kruskal–Wallis test was performed using the “stats” package in the R software [43]. The Correlation heatmap was plotted using the “corrmorant”, “pheatmap”, “ggplot2”, “ggcorrplot” and “corrgram” packages in the R software [44].

## 3. Results

### 3.1. Soil Properties

The soil chemical properties differed significantly between the manual plantation larch forest (mR) and the natural larch forest (NR) (Appendix A). In the plantation larch forest, the contents of soil AK and TK decreased, while the content of soil AN increased compared to the natural larch forest (Appendix A). However, most of the soil chemical properties, such as soil pH, TP, AP, and TN, did not change significantly (Appendix A).

### 3.2. Soil Fungal Diversity and Composition

The alpha diversity indices (i.e., Observed_species, Chao1, ACE, Shannon, and Simpson indices) of the soil fungal communities showed statistically significant differences between the two larch forest types (Appendix A). NMDS based on Bray–Curtis distances (stress < 0.1) showed that there were significant differences in the soil fungal community structure in the manual plantation larch forest (MR) and the natural larch forest (NR) (Figure 2).

The dominant fungal genera in the NR and MR soils were Mortierella, Russula, Pleotrichocladium, and Archaeorhizomyces (Figure 3a). However, when we used random forest classification models to predict the most important groups in the soil fungal communities, we found that some genera with relatively small abundances play a crucial role in fungal communities. The most important genera in the two forest soils were Humicola, Hymenoscyphus, and Lactarius (Figure 3b), which are relatively rare genera.

From the Kruskal–Wallis test, it was cleared that the abundance of some fungi is higher in natural larch forest than in manual plantation larch forest, e.g., Thelonectria, Saitozyma, Vytilinidion, and Hymenoscyphus. However, Suillus, Hygrophorus, and Varicosporium have a higher abundance in manual plantation larch forest than in natural larch forest (Figure 4). The Lefse cladogram analysis (Figure 4a) and the corresponding LDA score (Figure 5b) were conducted to examine which taxa (phylum to genus) changed in the different forest types (Figure 5). Figure 4 also displays the indicator species with LDA scores > 2. The genera Varicosporium and Geminibasidium showed abundance advantages in the natural larch forest (Figure 5a). At the same time, the genera Hymenoscyphus, Cylindrocarpon, Ilyonectria, Tomentella, Saitozyma, and Paratritirachium showed abundance advantages in the manual plantation larch forest (Figure 5a). Figure 4 shows that manual plantation larch forests have changed the indicator species of soil fungal communities.

From the size of the nodes in the heat tree, it can be seen that the relative abundance of Ascomycota at the phylum level and Dothideomycetes at the class level were the highest in all soil samples compared to other soil fungi (Figure 6). Additionally, we can conclude from the colour of each taxon that the log^−2^ ratio of median proportions is relatively high in Russulales, Sporormiaceae, and Cucurbitariaceae (Figure 6) for the same reason that the log^−2^ ratio of median proportions is relatively low in Atheliales, Chytridiomycota, and Filobasidiales (Figure 6).

### 3.3. Co-Occurrence Networks for Soil Fungal Taxa

We used the co-occurrence network analysis to investigate the interactions between the soil fungal taxa (Figure 7). Subsequently, we selected the top 300 genera in terms of abundance and performed a symbiotic network analysis based on a significance analysis (|r| > 0.6, *p* < 0.05). Visualisation of the soil fungal co-occurrence network demonstrated that the NR forest contained 1177 total links, with 99.66% positive and 0.34% negative links (Figure 7a). Compared to the NR forest, the complexity of the soil fungal co-occurrence networks in the MR forest decreased. There were 636 total links, with 97.8% positive and 2.2% negative links in the MR forest (Figure 7b). Moreover, the network topological characteristics (node number, total links, average degree, eigenvector centrality, average clustering coefficient, centralisation, connectedness, and relative modularity) also proved that the soil fungal co-occurrence network complexity decreased in the MR forest compared to the NR forest (Figure 7; Appendix A). According to the co-occurrence network, the top three ranked phyla (i.e., Ascomycota, Basidiomycota, and Mucoromycota) were found in the NR forest, while the top five ranked phyla (Ascomycota, Basidiomycota, Mucoromycota, Chytridiomycota, and Olpidiomycota) were found in the MR forest (Figure 7).

### 3.4. Factors Influencing the Soil Fungal Communities

The correlation heatmap showed that *Fusarium* had a significant negative correlation with TK; *Cenococcum* and *Archaeorhizomyces* had a significant negative correlation with TP; and *Humicolopsis* had a highly significant positive correlation with soil AN in natural larch forest (Figure 8a). In manual plantation larch forest, *Sebacina* and *Clavulina* had significant negative correlations with TP. *Cladophialophora* had significant negative correlations with TN, but *Lactarius* and *Cenococcum* had significant positive correlations with TN (Figure 8b). The Mantel test was used to examine the relationships of soil fungal community composition and the co-occurrence network of fungi with soil chemical properties (Figure 9). The complexity of the soil fungal co-occurrence network was related to the following contents of soil: AK, AN, TP, TK, AP, SOC, and pH (Figure 9). In contrast, the compositions of the fungal communities did not relate to the content of soil TP and AP (Figure 9).

The RDA indicated the relationship among the soil chemical properties, forest types, and relative abundances of fungal phyla (Figure 10). The RDA model explained 95.49% of the total variance. The AN was positively associated with Ascomycota, Mucoromycota, and Olpidiomycota. Chytridiomycota, Basidiomycota, and Zoopagomycota were positively associated with AP, TP, SOC, TK, AK, and pH.

### 3.5. Soil Fungal Community Assembly Processes

The null-model-based βNTI was used to assess the assembly processes of soil fungal communities in different forest types (Figure 11b). Soil fungal community assembly is influenced mainly by stochastic processes when the value of |βNT| is <2. According to the value of |βNT| in our results, the soil fungal community assembly processes of MR were mainly impacted by stochastic processes (Figure 11, |βNT| < 2), while the soil fungal community assembly processes of NR were impacted by deterministic processes (Figure 11, |βNT| > 2).

## 4. Discussion

### 4.1. Fungal Diversity and Composition under Manual Recovery

Soil fungal communities were known to be extremely sensitive and able to respond rapidly to changes in forest composition. They played a pivotal role in maintaining above-ground and below-ground linkages in forest ecosystems and in facilitating the flow and utilisation of nutrients in biogeochemical cycles. We observed significant differences in soil fungal alpha-diversity and β-diversity between manual plantation larch forest and natural larch forest (Figure 2; Appendix A). This may be due to the differences in dominant vegetation leading to changes in the soil environment and nutrient cycling, which in turn affects the adaptation of fungal communities [45,46]. Vegetation differences may also affect fungal interactions with other microorganisms, including competitive and symbiotic relationships, as well as triggering ecological niche differentiation and interspecific competition in fungal communities, ultimately leading to significant changes in soil fungal alpha-diversity [47,48]. Specifically, different vegetation affects soil physicochemical properties, such as pH and organic matter content, through its root secretions and detritus, providing different growth conditions for specific fungal populations [49]. At the same time, vegetation changes also altered nutrient cycling and energy flow patterns in the soil, prompting fungal communities to adapt to new environments.

The stack diagram indicated that the abundance of some dominant fungal genera undergoes a process of changes (Figure 3). The dominant genera, such as *Mortierella*, *Russula*, *Archaeorhizomyces*, and *Pleotrichocladium*, ranked as the top four in both forests (Figure 3a). However, compared to the natural larch forest, the proportion of *Archaeorhizomyces* increased and the proportion of *Pleotrichocladium* decreased in the Manual plantation larch forest (Figure 3a). The implementation of artificial vegetation restoration significantly affected the abundance of *Pleotrichocladium* and *Archaeorhizomyces* [50,51]. These two fungi showed high sensitivity to subtle changes in the soil environment, and dynamic changes in their abundance can serve as an effective bioindicator for assessing soil quality improvement [50,52]. This change is likely to be closely related to the alteration of the quantity and quality of decomposable carbon-containing substrates in the soil after vegetation restoration, which were an indispensable source of nutrients for normal growth and metabolic processes of the fungi. On the other hand, the organic matter content of the soil and its decomposition rate may have changed significantly with vegetation restoration, thus directly affecting the population dynamics of these sensitive fungi.

Notably, the random forest analysis results suggested that *Humicola* and *Hymenoscyphus* were key genera in soil fungal community under forest conversion (Figure 3b). These two fungal genera play an important role in maintaining the ecological balance of larch forests [50,53]. They construct a complex network of interactions with other soil microorganisms to maintain the stability and diversity of forest ecosystems, and play an indispensable role in the whole forest ecosystem. *Humicola* and *Hymenoscyphus* enhance the resilience of larch forests and help the trees to withstand unfavourable environmental conditions, such as drought and high and low temperatures [54,55]. This process not only promoted the growth of larch, but also improved the soil structure and created a better environment for other microorganisms, which was an important reason why they can be key genera.

According to the results of the Lefse, different biomarkers were found in both the manual plantation and natural larch forests. Varicosporium was a biomarker in the natural larch forest, and the genera Tomentella was a biomarker in the manual plantation larch forest (Figure 5a). It is known from previous reports that Tomentella is a dominant genus in the soil fungal community that has been identified as a key taxonomic genus in forest ecosystems. As a dominant genus, Tomentella has a significant and high correlation with other soil fungal genera (*p* < 0.05). Varicosporium is an indicator species in natural larch forests and there is no status for this indicator species in planted larch forests [56]. This may be related to the modification of the soil environment by the plantation forest, which makes the soil no longer suitable for Varicosporium [57]. The strong association between Varicosporium and plant diseases in natural forest ecosystems, sometimes leading to plant death, demonstrates its important role in maintaining the ecological balance of natural forests [58]. We hypothesised that in natural planted larch forests, Varicosporium may have a symbiotic or parasitic relationship with larch roots or be involved in key ecological processes such as decomposition of organic matter. These speculations are based on its close relationship with plant diseases. However, in planted larch forests, the status of Varicosporium has changed significantly due to changes in the soil environment.

### 4.2. Soil Fungal Co-Occurrence Networks under Different Forest Types

The results of the soil fungal co-occurrence network analysis demonstrated that the complexity and connection of soil fungal communities have changed, and the stability and resistance of soil fungal communities in manual plantation larch forests have decreased (Figure 7), which is consistent with previous studies. Wu et al. (2023) reported that the complexity of the soil fungal co-occurrence network has decreased in artificial larch forests, and the number of nodes and edges in the symbiotic network has also decreased [59]. Similar to previous studies, Wang et al. (2023) indicated that the complexity of soil fungal co-occurrence networks in plantation forests is reduced compared to that of natural forests [60]. Moreover, the artificial restoration of forest ecosystems in small areas around cities has shown a reduction in the complexity of fungal co-occurrence networks [61]. We suggest that the reason for the lower complexity in manual plantation larch forests may involve several aspects. Firstly, the soil fungal communities in natural larch forests form complex interactive relationships with different taxa after long-term natural evolution and adaptation, and such relationships help to construct complex soil fungal co-occurrence networks. Secondly, soil environments and nutrient conditions may also differ between manual plantation larch forests and natural larch forests. Soils in natural larch forests usually undergo long-term natural succession and accumulation, and they are rich in organic matter and trace elements, providing soil fungi with abundant survival resources and habitats. In contrast, soils in manual plantation larch forests may be planted with only a single plant species, leaving the soil low in nutrients and limiting the complexity of the co-occurrence network of soil fungi.

From the soil fungal co-occurrence network, we noticed that Chytridiomycota and Olpidiomycota were the emerging fungal phyla among the top five dominant phyla (*p* < 0.05). Based on the previous studies, Chytridiomycota is relatively abundant in freshwater and saline habitats and also significantly present in soil [62]. The emergence and high ranking of Chytridiomycota and Olpidiomycota may be due to the fact that the plants introduced into manual plantation larch forests are more sensitive to pathogenic attack, and this is also the reason why Chytridiomycota and Olpidiomycota are among the top five dominant phyla. Soil nutrients have important effects on the growth and distribution of different soil fungi. There may be differences in soil nutrients, such as organic matter content and moisture conditions, between manual plantation and natural larch forests. These differences may affect the growth and reproduction of Chytridiomycota and Olpidiomycota and lead to an increase in their proportion in manual plantation larch forests.

Differences in fungal communities and network complexity have profound effects on ecosystems [63]. They not only reduce the threat of invasive species by improving ecosystem stability and resistance to disturbance, but also play a key role in nutrient cycling, influencing nutrient flow and utilisation efficiency [64]. At the same time, fungi promote plant growth by stabilising soil structure and maintaining soil aeration, permeability and water retention properties, which in turn promote plant growth. In addition, as part of biodiversity, fungi live in symbiosis with a wide range of species, providing rich resources and ecological niches for ecosystems, thus affecting the level of biodiversity in the whole ecosystem. Thus, differences in fungal communities and network complexity work together at multiple levels to contribute to the overall functioning and health of ecosystems.

### 4.3. Factors Influencing Soil Fungal Communities in Two Forest Types

The correlation heatmap indicated that *Cenococcum* and *Archaeorhizomyces* and soil TP were significantly and negatively correlated in natural larch forests. In contrast, *Lactarius* and *Cenococcum* had a significant positive correlation with soil TP in manual plantation larch forest (Figure 8). Considering the differences in correlations between soil total phosphorus (TP) and fungal genera in natural and planted larch forests, one possible reason is the different availability and cycling mechanisms of soil nutrients in the two forest types. In natural larch forests, soil phosphorus may be more immobilised in organic matter or soil minerals and not readily available for direct microbial utilisation. Therefore, fungi like *Cenococcum* and *Archaeorhizomyces* may have a competitive relationship with soil phosphorus, resulting in a negative correlation between them and soil TP [51,65]. This may be because these fungi release enzymes such as phosphatases as they decompose organic matter in an attempt to convert organic phosphorus into utilisable inorganic phosphorus, but in the process they also reduce the amount of directly measurable TP in the soil. In contrast, in planted larch forests, the effectiveness of phosphorus in the soil may have increased, allowing some fungi such as *Lactarius* and *Cenococcum* to more readily utilise these nutrients [66,67]. These fungi may help plants to absorb soil phosphorus through a symbiotic relationship with plant roots, which in turn promotes the transformation and cycling of soil TP, thus showing a positive correlation with soil TP.

Moreover, the results of the Mantel test showed that the complexity of the soil fungal network was linked to various soil contents such as AK, AN, TP, TK, AP, SOC, and pH (Figure 9). We speculated that this may be due to the fact that the chemical constituents lying in the soil such as fast-acting potassium, fast-acting nitrogen and soil organic carbon provide the necessary nutrients and energy sources for the fungi, which directly affects their metabolic activities and population diversity [68]. Meanwhile, soil pH, as an important environmental factor, significantly affects the growth environment of fungi and their enzyme activities [69,70]. Although total phosphorus and total potassium are not directly utilised by fungi, they serve as important indicators of soil fertility and also indirectly reflect the advantages and disadvantages of the fungal growth environment. These chemical compositions and pH act together on fungi through multiple mechanisms to affect their growth, reproduction and metabolism, thus shaping a complex and diverse soil fungal network.

The RDA analysis showed that Chytridiomycota, Basidiomycota, and Zoopagomycota were positively associated with AP, TP, SOC, TK, AK, and pH. Ascomycota, Mucoromycota, and Olpidiomycota were positively related to AN (Figure 10). It is well known that soil organic carbon (SOC) provides Chytridiomycota, Basidiomycota, and Zoopagomycota with a source of energy and a carbon skeleton for building cellular structures [71,72]. In turn, these fungi obtain energy by decomposing and utilising organic carbon to support their life activities and growth. Soil fungal taxa such as Chytridiomycota, Basidiomycota, and Zoopagomycota may be more adapted to nutrient-rich, moderately acidic and alkaline soil environments [73,74,75], and, therefore, positively correlated with these chemical indicators. In addition, fungi such as Ascomycota, Mucoromycota, and Olpidiomycota may possess special enzyme systems that can efficiently break down organic nitrogen in the soil, such as proteins and amino acids, into forms that can be directly absorbed and utilised by the fungi, such as ammoniacal nitrogen or nitrate nitrogen. This allowed them to grow and multiply rapidly in fast-acting nitrogen-rich environments.

### 4.4. Assembly of Fungal Community under Manual Recovery

Our results indicated that deterministic processes mainly influenced soil fungal community assembly in natural larch forests, whereas stochastic processes mainly influenced soil fungal community assembly in manual plantation larch forests (Figure 11). This suggested that the soil fungal communities in the natural larch forest were more susceptible to deterministic factors in the surrounding environment (selection by abiotic environmental factors and symbiotic and antagonistic interactions between species), whereas the soil fungal communities in the manual plantation larch forest were less susceptible to these deterministic factors and were, therefore, more inclined to respond to stochastic events.

The study of Huo et al. (2023) showed that, in forest ecosystems, the soil fungal community assembly was mainly influenced by a stochastic process (dispersion-constrained) [24]. However, Chen et al. (2023) reported that the soil fungal community assembly was mainly influenced by deterministic processes in artificial recovery forests in subtropical forest ecosystems [76]. Additionally, Lan et al. (2022) demonstrated that the soil fungal community assembly in tropical forests was dominated by stochastic processes [77]. The reason for these differences may be due to the differences in soil nutrient content, pH, and moisture conditions in forest ecosystems in different geographic locations, in turn affecting the soil fungal community assembly. Whereas, in our results, the soil fungal assembly in the manual plantation larch forest was dominated by stochastic processes, and that of the natural larch forest was dominated by deterministic processes. We hypothesised that this phenomenon may be caused by the differences in the soil environments between manual plantation and natural larch forests and the resulting changes in soil fungal interactions. Manual plantation larch forests are usually planted with a single or few tree species, resulting in relatively low vegetation diversity, whereas natural larch forests are characterised by an abundance of plant species and complex ecosystem interactions. As a result of changes in vegetation and soil conditions, fungal interactions such as competition and symbiosis may change, which, in turn, affects soil fungal community assembly.

There are still some limitations in this study. Firstly, in terms of soil sampling, there may be cases where the sample collection points are not well chosen, such as choosing to sample places that are contaminated or affected by other sources of contamination, which will lead to deviation of the assay results from the actual situation. In addition, in terms of sequencing methods, although high-throughput sequencing technology has greatly improved the output and sequencing speed, the sequencing error rate still exists, which may affect the accuracy of the results. Meanwhile, data processing is also a complex process that requires solving problems in data transmission, processing and storage. Finally, the interpretation of sequencing results is also a challenge, as it requires the development of specialised analytical tools and algorithms and an in-depth understanding of the interactions between biological processes and DNA sequences. These limitations suggest that we need to be more cautious when conducting similar studies and take measures to minimise potential bias.

## 5. Conclusions

Our study in Heilongjiang Zhongyangzhan Black-billed Capercaillie Nature Reserve and Gala Mountain Forest provides valuable insights into the effects of soil fungal communities under natural larch forests and manual plantation larch forests. Our results showed that there was a significant difference in the soil fungal communities’ alpha diversity, and there were some differences in the soil fungal communities’ structures, as well as changes in the abundance of some dominant genera in the soil fungal communities in the two different forest types. The Kruskal–Wallis test showed that different genera of fungi differed somewhat under different forest types. However, analyses using the random forest model highlighted the key role played by some rare soil fungi, which are often overlooked in traditional studies. In addition, an important finding of our study is the reduced complexity of soil fungal co-occurrence networks in manual plantation larch forests compared to natural larch forests. This suggested that the stability and resilience of the soil fungal community decreased. Our study revealed a dynamic balance between stochastic and deterministic processes in soil fungal community assembly. In manual plantation larch forests, soil fungal community assembly was mainly influenced by stochastic processes. Notably, this study analysed the changes in soil chemical properties, soil fungal community diversity and composition, soil fungal community co-occurrence networks, and soil fungal community assembly processes in different forest types. The above analyses help us to better understand the ecological changes in soil fungal communities in both manual plantation and natural larch forests. These findings emphasise the centrality of soil fungi in forest ecosystems, where changes in community structure and diversity have far-reaching implications for ecosystem health and stability. Future studies should deeply investigate the long-term dynamic changes of soil fungal communities in planted and natural forests and their impacts on forest ecosystem functions and services, in order to further reveal the key role of soil fungi in maintaining forest health and stability.

## Figures and Tables

**Figure 1 microorganisms-12-01322-f001:**
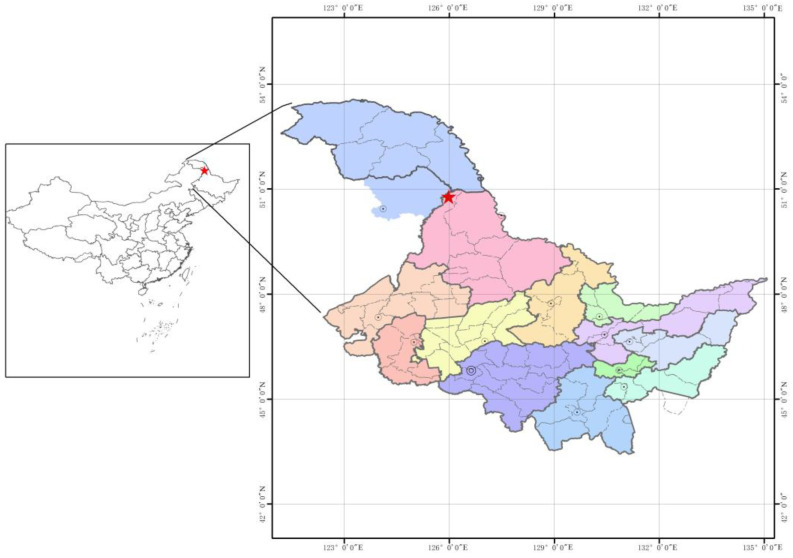
The study site is located in Heilongjiang Province, China. The red stars in the image represent the sampling location sites.

**Figure 2 microorganisms-12-01322-f002:**
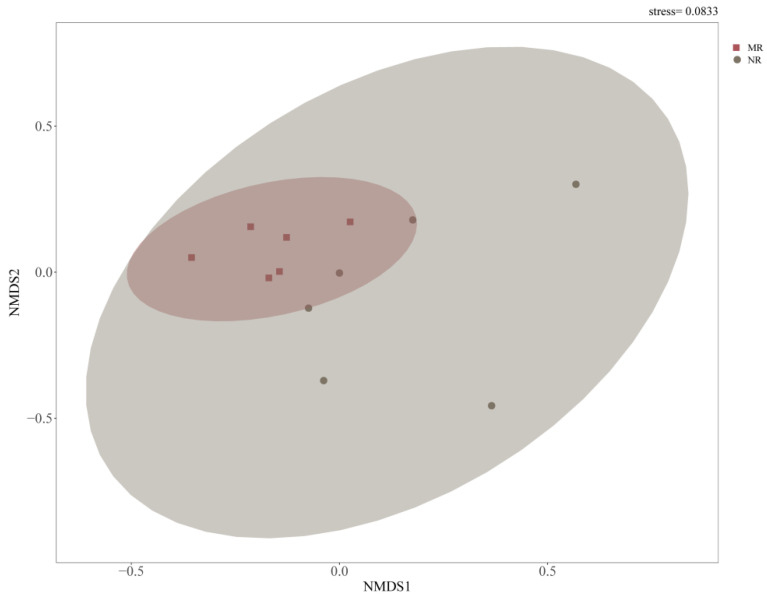
Nonmetric multidimensional scaling (NMDS) analyses of fungal β-diversity based on Bray–Curtis distance metrics. Different coloured dots represent different forest types. The stress value represents a measure of the error between the original distance and the low-dimensional spatial distance obtained using NMDS. MR: manual plantation larch forests, NR: natural larch forests.

**Figure 3 microorganisms-12-01322-f003:**
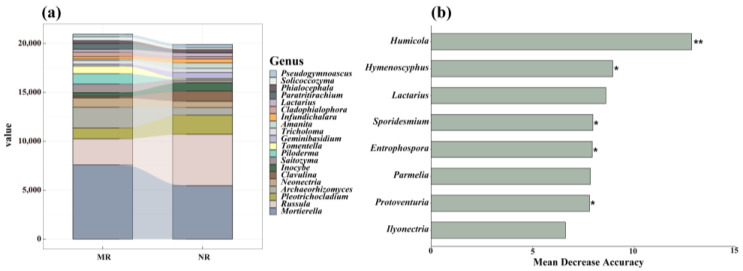
The top-20 soil fungal genera in the two forest types (MR, NR) are displayed in (**a**). The random forest model showed the key genera that play an important role in larch forests (**b**). An asterisk near each bar indicates whether each predictor is significant. MR: manual plantation larch forests, NR: natural larch forests. ‘*’ and ‘**’ indicate significance. (‘*’ indicates *p* < 0.05; ‘**’ indicates *p* < 0.01).

**Figure 4 microorganisms-12-01322-f004:**
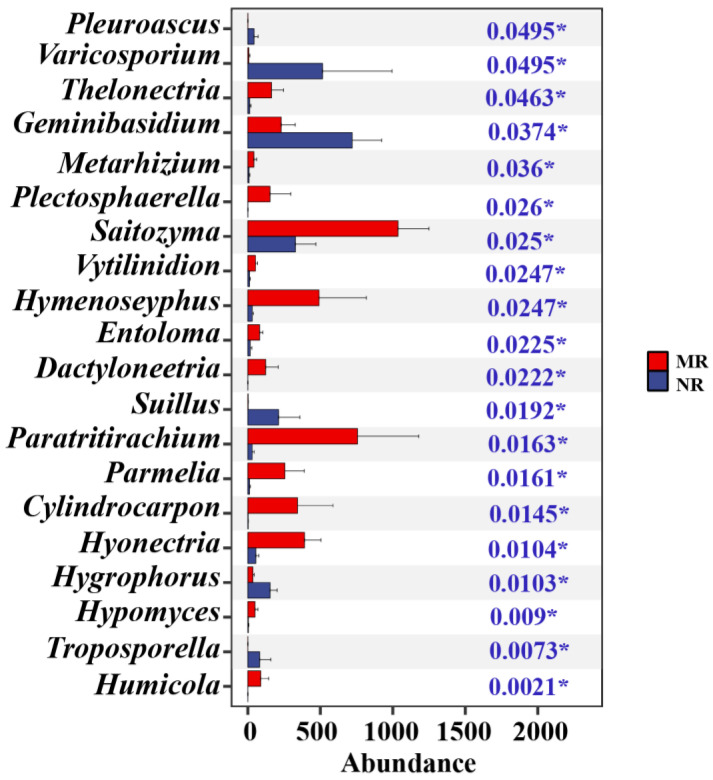
The Kruskal–Wallis test was used to determine the differences between the soil fungal genera of the two forest types. (‘*’ indicates *p* < 0.05). The blue numbers (*p*-values) on the right side of the graph are used to determine if there is a significant difference between the groups of data.

**Figure 5 microorganisms-12-01322-f005:**
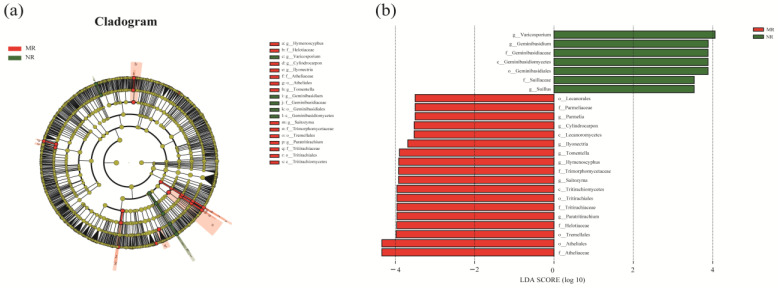
Lefse analysis based on OTUs in different forest types (NR, MR). The circles radiating from inside to outside represent taxonomic levels from phylum to genus (or species) (**a**). The legend shows the names of the species indicated by the letters in the figure. LDA threshold: only species that differed above this threshold with *p* < 0.05 were considered marker species. The distribution bar chart mainly shows the significantly different species with LDA scores greater than the preset value (**b**), namely, the biomarker with statistical differences, with a preset value of 2.0. MR: manual plantation larch forests, NR: natural larch forests.

**Figure 6 microorganisms-12-01322-f006:**
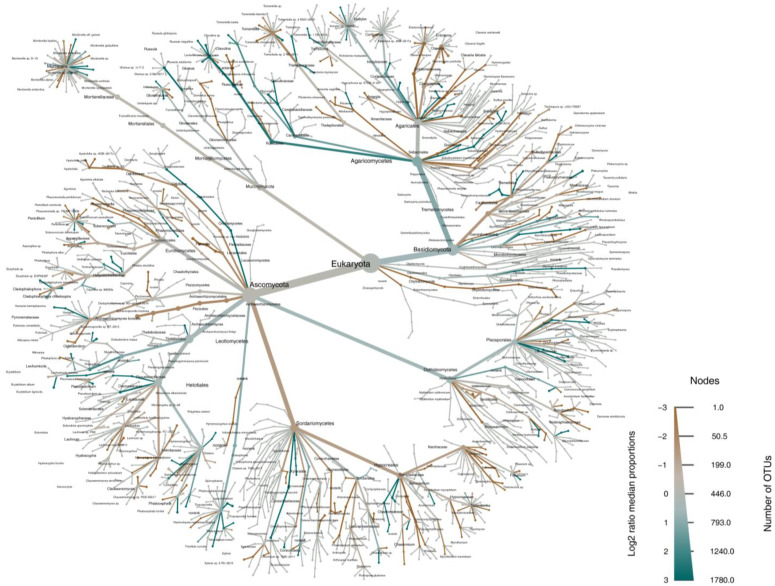
Heat trees displaying the average proportions of soil fungi in forests. Nodes represent each taxonomic rank from kingdom (fungi, centre) to species (tips of each branch). Node and edge (branch) width indicates the mean proportion of that taxon in samples belonging to that group. The size of the nodes corresponds to the number of taxa. Only significant differences are coloured, determined using a Wilcoxon rank-sum test followed by a Benjamini–Hochberg (FDR) correction for multiple comparisons.

**Figure 7 microorganisms-12-01322-f007:**
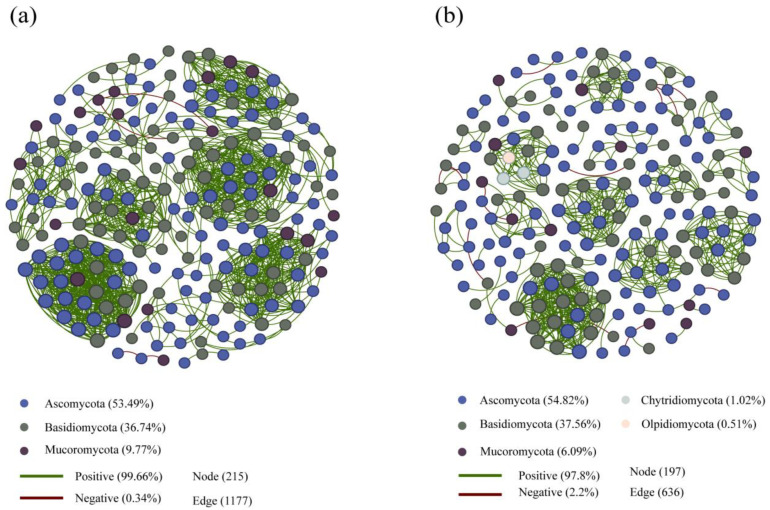
The co-occurrence network shows the complexity between soil fungi for NR (**a**) (n = 300) and MR (**b**) (n = 300). The connecting lines represent strong and significant correlations (*p* < 0.05). Different dots represent different genera. Different coloured lines indicate different correlations. Deep-blue lines represent positive effects and light-blue lines represent negative effects. MR: manual plantation larch forests, NR: natural larch forests.

**Figure 8 microorganisms-12-01322-f008:**
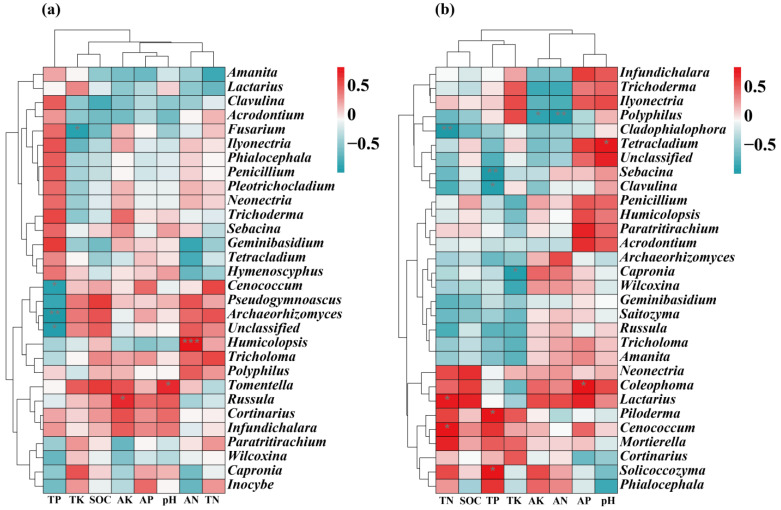
Correlation heatmap showing the correlation between the top ranked fungal genera and soil physico-chemical properties in NR (**a**) and MR (**b**). Red represents positive correlation and green represents negative correlation. ‘*’ indicates *p* < 0.05; ‘**’ indicates *p* < 0.01; ‘***’ indicates *p* < 0.001). The correlation between different variables was derived based on the Pearson correlation coefficient (Pearson correlation coefficient).

**Figure 9 microorganisms-12-01322-f009:**
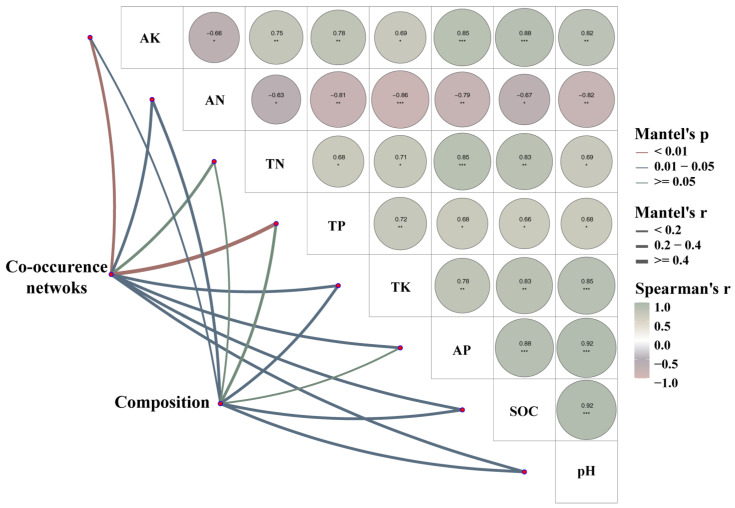
Relationship of soil fungal community composition and co-occurrence networks with soil chemical properties. The red and blue lines represent different levels of correlation, and the green line represents no correlation. The thickness of the line (Spearman’s correlation coefficients) represents the magnitude of the correlation. The thicker the line, the greater the correlation. AK: available potassium. AN: available nitrogen. TN: total nitrogen. TP: total phosphorus. TK: total potassium. AP: available phosphorus. SOC: soil organic carbon. Composition: microbial Bray–Curtis dissimilarity. (‘*’ indicates *p* < 0.05; ‘**’ indicates *p* < 0.01; ‘***’ indicates *p* < 0.001).

**Figure 10 microorganisms-12-01322-f010:**
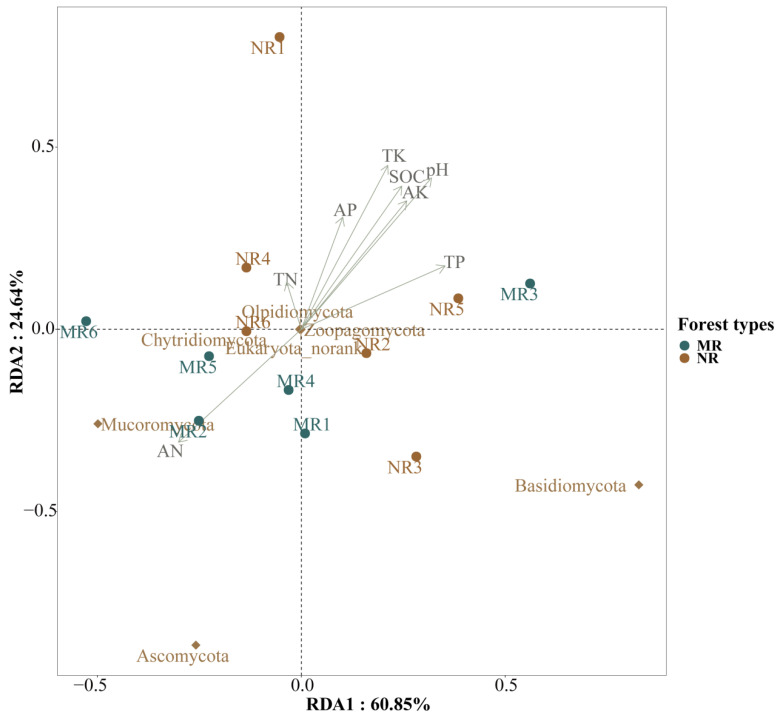
Redundancy analysis (RDA) of the relationships in soil fungal communities and soil chemical properties under different forest types. The closer the distance between two dots, the higher the functional similarity of the two samples. The longer the ray, the greater the influence of the factor on the structure and function of the colony; the angle between the arrow ray and the coordinate axis represents the size of the correlation between a certain environmental factor and the coordinate axis; the smaller the angle, the higher the correlation; the position of the sample projection point on the blue arrow: an approximate representation of the size of the value of the factor in the corresponding sample. Percentage next to the axes represents the proportion of the variance in the raw data that can be explained by the corresponding axes. MR: manual plantation larch forests, NR: natural larch forests.

**Figure 11 microorganisms-12-01322-f011:**
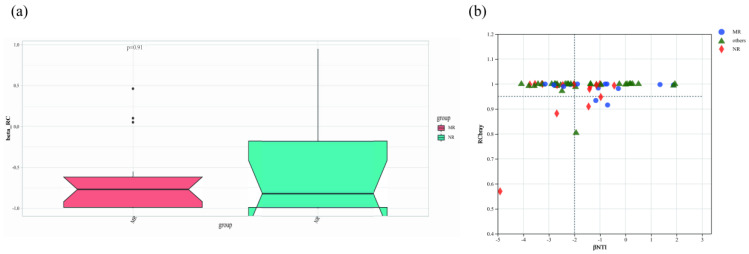
The null model analysis to explain the community assembly mechanisms of soil fungal communities in different forest types. RCbray–Curtis > 0.95 implies diffusion limitation (probabilistic diffusion), RCbray–Curtis < 0.95 implies homogeneous diffusion (homogeneous diffusion) and −0.95 < RCbray–Curtis < 0.95 implies drift (undominated process). MR: manual plantation larch forests; NR: natural larch forests.

## Data Availability

Data are contained within the article.

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
