# Peer review of "Soil Fungal Community Differences in Manual Plantation Larch Forest and Natural Larch Forest in Northeast China"

_microorganisms, 2024, doi:10.3390/microorganisms12071322_

Round 1

Reviewer 1 Report

Comments and Suggestions for Authors

It is well known that under red pine forests over time very humus-rich / wintery/ soils with outstanding fertility are formed, which gardeners are happy to utilize.

The thesis presents in detail the fungal community of a planted and a natural red pine forest soil, then comparing these, analyses microevolutionary changes from several aspects, advantageously exploiting the possibilities offered by the new generation sequencing methods.

It analyses the structure and diversity of the community in several ways and its consequences for ecosystem health and stability.

The manuscript also thoroughly analyses the soil needs of the members of the mushroom community.

The literature is also well developed, and the high proportion of recent papers is commendable.

Omissions, incorrect editing or translation:

ad fig. 3.b. no asterisks

ad fig 4 (l. 245.): There is no need for: "'**' indicates P < 0.001)"

AD 287. "The Top Ranked Bacterial Genera"?

AD 289: Asterisks?

Formal errors.AD 123-128: Writing compounds incorrectly

Genus and species names should be italicized (Italic): 370, 371, 274, 375, 372, 378, 379, 381, 384, 

569, 571, 572, 651, 653, 661, 662, 666-668, 683

Incomplete reference: 710-711.

Comments on the Quality of English Language

The use of English corresponds to the quality expected for scientific papers.

Reviewer 2 Report

Comments and Suggestions for Authors

In the present manuscript titled" Soil Fungal Communities Difference in Manual Plantational Larch Forest and Natural Larch-forests in Northeast China ", the authors performed a study to provide new information on the variation of soil fungal communities between natural larch forests and manual plantation larch forests in the Heilongjiang Zhongyangzhan Black-beaked Grouse Nature Reserve using high-throughput sequencing technology to determine the structure and composition of fungal communities.

The results showed that there was no significant change in the alpha diversity of the soil fungal community, but only a significant change in the β diversity of the soil fungal communities between natural larch forests and manually planted larch forests.

A further indication is provided by the fact that the soil fungal communities in the manually planted larch forest was mainly driven by stochastic processes, suggesting that the fungal communities were adaptive to environmental changes.

The application of Mantel and RDA analisys to detect correlation and influences between soil fungal co-occurence network, soil fungal community and environmental parameters are reliable and effective

These results, which are experimentally reliable, leave little room for comment and allow acceptance of the rational conclusions reached by the authors.

Therefore, this work can be accepted if the authors emphasize in the introduction the possible interrelation  between fungus and root that may affect colony development.

To this end, I suggest inserting at line 65 : Given this background, investigating the community assembly process of soil fungi and its co-occurence network in manual plantation larch forest and natural larch forest is crucial for understanding the construction mechanism and ecological adaptations of fungal communities, which not only helps to reveal the functions and interactions of soil fungi in ecosystems, but also provides a powerful guide for biodiversity  conservation and ecological restoration to which the interrelationship between fungus and root also contributes, which can affect colony development in different habitats and nutrient conditions.(Di Martino et al 2022 Mycorrhized Wheat Plants and Nitrogen Assimilation in Coexistence and Antagonism with Spontaneous Colonization of Pathogenic and Saprophytic Fungi in a Soil of Low Fertilityhttps://doi.org/10.3390/plants11070924) 

Comments on the Quality of English Language

Minor editing of English language required

Reviewer 3 Report

Comments and Suggestions for Authors

General comments

Introduction

Could you provide more context on the ecological importance of larch forests specifically in Northeast China?

How do the roles of soil fungi in nutrient cycling and symbiosis with plants differ in natural versus plantation forests?

Can you clarify the significance of studying soil fungal communities in the context of biodiversity conservation and ecological restoration?

Materials and Methods

Can you provide more detailed information on the differences between the natural and manual plantation sites, such as historical land use or management practices?

Were there any controls or replicates used in the soil sampling? If so, could you specify the details?

What measures were taken to ensure the accuracy and reproducibility of the DNA extraction and sequencing processes?

Discussion

How do your findings compare to previous studies on soil fungal communities in natural and plantation forests?

What are the potential ecological implications of the observed differences in fungal communities and network complexities?

Can you discuss any limitations of the study, such as potential biases in soil sampling or sequencing methods?

What future research directions do you suggest based on your findings?

Comments on the Quality of English Language

Round 2

Reviewer 3 Report

Comments and Suggestions for Authors

I am pleased to inform you that the authors have thoroughly addressed all the comments and suggestions I provided for their manuscript entitled “Soil Fungal Community Differences in Manual Plantation Larch Forest and Natural Larch Forest in Northeast China (microorganisms-3050615).” The revisions have significantly improved the quality of the manuscript.

The authors have provided detailed and thoughtful responses to each of my comments and have made the necessary revisions throughout the manuscript to address the concerns raised.